# Overexpression of E-Cadherin Is a Favorable Prognostic Biomarker in Oral Squamous Cell Carcinoma: A Systematic Review and Meta-Analysis

**DOI:** 10.3390/biology12020239

**Published:** 2023-02-02

**Authors:** Alejandro I. Lorenzo-Pouso, Fábio França-Vieira e Silva, Alba Pérez-Jardón, Cintia M. Chamorro-Petronacci, Mônica G. Oliveira-Alves, Óscar Álvarez-Calderón-Iglesias, Vito Carlo Alberto Caponio, Morena Pinti, Vittoria Perrotti, Mario Pérez-Sayáns

**Affiliations:** 1Oral Medicine, Oral Surgery and Implantology Unit (MedOralRes), Faculty of Medicine and Dentistry, Universidade de Santiago de Compostela, 15782 Santiago de Compostela, Spain; 2ORALRES Group, Health Research Institute of Santiago de Compostela (FIDIS), 15782 Santiago de Compostela, Spain; 3Technology Research Center (NPT), Universidade Mogi das Cruzes, Mogi das Cruzes 12245-000, SP, Brazil; 4School of Medicine, Anhembi Morumbi University, São José dos Campos 12247-004, SP, Brazil; 5Research, Health and Podiatry Group, Department of Health Sciences, Faculty of Nursing and Podiatry, University of A Coruña, 15008 A Coruña, Spain; 6HM Hospitals Research Foundation, 28015 Madrid, Spain; 7Department of Clinical and Experimental Medicine, University of Foggia, 71122 Foggia, Italy; 8Department of Medical, Oral and Biotechnological Sciences, University “G. d’Annunzio” Chieti-Pescara, 66100 Chieti, Italy

**Keywords:** E-cadherin, epithelial-mesenchymal transition, mouth neoplasm, prognostic, systematic review, meta-analysis

## Abstract

**Simple Summary:**

An oral cavity tumor, known as an oral squamous cell carcinoma (OSCC), has a five-year survival rate of just about 50%. Novel, easily, available biomarkers for prognosis evaluation are still required. Despite advancements in diagnosis and therapy, this rate has not grown in recent decades. Our study indicated that a lower expression of a protein named E-cadherin is related with a poor prognosis for OSCC patients. This evidence came from the meta-analysis of 25 studies, including 2553 patients, in which E-cadherin protein expression was assessed in their tumor sample by immunohistochemistry. This study highlights the promising role of E-cadherin assessment during routine histopathology diagnoses to support prognostic decision-making, and to pave the way for future studies to counteract its role in cancer progression.

**Abstract:**

Oral squamous cell carcinoma (OSCC) is characterized by poor survival, mostly due to local invasion, loco-regional recurrence, and metastasis. Given that the weakening of cell-to-cell adhesion is a feature associated with the migration and invasion of cancer cells, different studies have explored the prognostic utility of cell adhesion molecules such as E-cadherin (E-cad). This study aims to summarize current evidence in a meta-analysis, focusing on the prognostic role of E-cad in OSCC. To find studies meeting inclusion criteria, Scopus, Web of Science, EMBASE, Medline, and OpenGrey databases were systematically assessed and screened. The selection process led to 25 studies, which were considered eligible for inclusion in the meta-analysis, representing a sample of 2553 patients. E-cad overexpression was strongly associated with longer overall survival (OS) with Hazard Ratio (HR)  = 0.41 95% confidence interval (95% CI) (0.32–0.54); *p* < 0.001 and disease-free survival with HR 0.47 95% CI (0.37–0.61); *p* < 0.001. In terms of OS, patients with tongue cancer experienced better survivability when expressing E-cad with HR 0.28 95% CI (0.19–0.43); *p* < 0.001. Globally, our findings indicate the prognostic role of the immunohistochemical assessment of E-cad in OSCC and its expression might acquire a different role based on the oral cavity subsites.

## 1. Introduction

Worldwide, oral squamous cell carcinoma (OSCC) is a growing health problem. The prognosis for patients affected by this neoplasm remains mainly based on the TNM system [1]. Therapeutics have improved during recent decades, although these efforts remain suboptimal, owing to the remaining 5-year survival rate of around 50–60%. These long-term outcomes are even poorer in tumors at advanced stages, mainly due to poor locoregional control and the increased odds of distant metastasis development [2]. Recent initiatives have addressed the identification of biomarkers to complement prognosis assessment from a molecular point-of-view to study biological tumor behavior, but also in the search for novel therapeutic targets [3,4,5]. In the last decade, the number of studies that focused on the prognostic value of immunohistochemical (IHC) biomarkers has increased. Among these, one of the most studied biological mechanisms underlying OSCC is epithelial–mesenchymal transition (EMT) [6]. EMT includes epithelial cells showing mesenchymal traits and characteristics [7]. This EMT-related phenotype switches results in a cascade of aberrant signaling pathways such as TFG-β, Wnt signaling or notch, and aberrant stemness properties [8,9]. Specially, EMT is mainly driven by a molecular machinery of glycoproteins that mediate intercellular adhesion via calcium-dependent binding; among these, the main family of proteins involved are cadherins [10,11,12]. The protein product of the gene CDH1 is one of the most studied in OSCC and EMT. The CDH1 gene is in the 16q22.1 chromosomal band and its protein product is E-Cadherin (E-cad). E-cad is important in maintaining oral epithelial cell homeostasis and physiology [13]. Different studies showed that E-cad inactivation triggers a decrease in cell–cell contact, followed by the loss of cell polarity, differentiation, and a gain in growth and cell migration (Figure 1) [14,15]. 

E-cad is a well-characterized molecular marker of EMT, and changes in its expression have been related to the detrimental prognosis of several solid tumors [16]. Association between reduced E-cad expression and poor survival was found in a previously published meta-analysis in head and neck [17], esophageal [18], and gastric cancer [19]. Several molecular processes contribute to decreased E-cad expression, linked to the onset and progression of different cancers. The promoter hypermethylation, a crucial tumor suppressor–silencing mechanism, has been linked significantly to decreased E-cad expression [20]. Furthermore, reduced expression of E-cad can be a consequence of somatic and germline mutations [21]. Additionally, new research has demonstrated that downregulation occurs as a result of the activation of E-cad transcriptional repressors such as Snail and Slug [22,23]. Several microRNAs, including those in the miR-200 family and miR-101, have been found to suppress E-cad by upregulating the expression of E-cad repressors when their expression is decreased or deleted [24,25]. These mechanisms differentially contribute to lower E-cad expression, leading to cancer onset and progression.

Nonetheless, the association between E-cad IHC-based expression and long-term outcomes has proved controversial according to previous meta-analyses regarding head and neck squamous cell carcinomas [17]. The only meta-analysis carried out specifically in OSCC presented several drawbacks, such as the exclusion of subgroup analysis. 

Therefore, the aim of this systematic review and meta-analysis was to further investigate the potential role of E-cad immunohistochemical expression to serve as a prognostic factor for long-term outcomes related to survival in OSCC.

## 2. Materials and Methods

Before performing the systematic review, its protocol was registered in the PROSPERO database (CRD42020201631). This systematic review and meta-analysis comply with PRISMA [26] and MOOSE guidelines [27]. 

The research question was developed according to the PICO framework: What is the prognostic significance of E-Cadherin immunohistochemical expression in patients with OSCC? This is in accordance with the PICO method: population (adults), intervention (E-cadherin overexpression), comparison (E-cadherin subexpression), outcome (long-term outcomes/survival).

### 2.1. Search Strategy

References were screened from different databases. The Web of Science, Scopus, Medline via PubMed, EMBASE, and Grey Literature Database were accessed (accessed on 30 July 2022). Searches were conducted by combining MeSH and EMTREE terms used by the databases and general text words. The following algorithm was applied to Medline: (“E-cadherin” [MeSH Terms]) OR (“cadherin” [All Fields] AND “E-cad” [All Fields]) OR (“epithelial cadherin” [All Fields]) AND (“mouth” [MeSH Terms] OR “mouth” [All Fields] OR “oral” [All Fields]) AND (“carcinoma, squamous cell” [MeSH Terms]) OR (“carcinoma” [All Fields] AND “squamous” [All Fields] AND “cell” [All Fields]) OR (“squamous cell carcinoma” [All Fields]). This syntax was conveniently adapted for each database.

A supplemental manual search was also conducted in journals relating to oral pathology, oral/maxillofacial surgery, oral medicine, and oncology. Reference lists from the retrieved papers as well as potentially pertinent articles that any of the writers were familiar with were thoroughly verified. All references retrieved were managed using the software EndNote X9 (Clarivate, PA, USA), and duplicated references were also deleted with this software.

### 2.2. Eligibility Criteria

An ad hoc review group was composed by two experts in oral pathology (AILP and MPS). The study selection was carried out in two screening phases. First, these two authors independently evaluated titles and abstracts of resulting studies, further solving disagreements in a consensus meeting. Subsequently, full-text studies were blindly appraised by the same authors and the information was again cross-checked. Freeware Epidat 4.2 (Software for Epidemiological Analysis of Tabulated Data. www.sergas.es/saude-publica/EPIDAT accessed on 30 July 2022) was used to calculate Cohen’s kappa coefficient (κ) as a measure of inter-observer agreement. A third researcher was involved to solve disagreements in each stage (CMCP).

Inclusion criteria: (i) original research studies published in English; (ii) evaluation of E-cad expression in human primary OSCC tissue by IHC; (iii) E-cad overexpression association analysis including E-cad with at least one long-term outcome: overall survival (OS), disease-free survival (DFS), or disease-specific survival (DSS). Lacking standardization in survival endpoints nomenclature, studies evaluating the above-mentioned terms or related terms that match these definitions (e.g., progression-free survival) were included in each endpoint category.

Exclusion criteria: (i) reviews, meta-analysis, case reports, clinical trials, and editorials; (ii) in vitro or animal-based studies; (iii) genomic-based research; (iv) studies that do not deal with the relationship between E-cad expression and time-dependent outcomes; (v) or in which presented data are insufficient to estimate Hazard Ratios; (vi) studies with duplicated cohorts.

### 2.3. Data Extraction

The first author’s name, the year of publication, and the country and continent where the study was conducted were included in the data extraction sheet. This information was matched with the total number of patients included; edition for staging OSCC; recruitment period; tumor subsite localization in the oral cavity; treatment modality; follow-up period; cut-off value for E-cad IHC positivity and if there was a nuclear or cytoplasmic immunostaining pattern; hazard ratios (HRs) for long-term outcomes and covariates for multivariate analysis (i.e., Cox regression models).

### 2.4. Quality Assessment

Included studies underwent a risk of bias assessment and checked for items included in the Reporting Recommendations for Tumor Marker Prognostic Studies (REMARK) guidelines [28]. Each evaluated item and criteria are already previously reported and explained [29]. APJ and MPS independently scored all the included studies. Again, in case of disagreements, a third reviewer participated (AILP) until a consensus was reached. Studies with an overall score >4 were considered at a low risk of bias.

### 2.5. Statistical Analysis

Based on the original papers’ cut-off, patients were categorized as having high or low levels of E-cad expression. To estimate the impact of E-cad expression on time-dependent outcomes, such as OS, DSS, and DFS, HRs and 95% confidential intervals (CIs) from both univariate and multivariate analyses were used. For each study, we used the estimate that was adjusted for the largest number of variables. In case HRs were not reported in the original manuscript, their values were extracted from Kaplan–Meier curves as described by Parmar et al. [30] and Tierney et al. [31] using Engauge Digitizer 4.1 (open-source digitizing software developed by M. Mitchell). The corresponding or first authors were also emailed when it was not possible to estimate the HRs as described above. 

Weighted adjusted log HRs by the inverse of their variance were pooled in the meta-analysis to calculate a pooled HR and its 95% CIs. Pooled analysis was performed using both the fixed-effect method and random-effects models. The estimates considered as most appropriate figures were selected based on between-study heterogeneity. Forest plots were created to graphically represent the pooled analysis. A subgroup analysis based on several variables was planned (i.e., tumor subsite, quality score, E-cad antibody, ethnic variations, type of covariate adjustment, and cut-off point) to further explore the origin of heterogeneity. The statistical significance was fixed at α = 0.05.

The Cochran’s Q statistic test, which is based on I^2^ statistics and the Chi-squared test, was used to evaluate heterogeneity. For the Cochran’s Q test, heterogeneity was deemed significant if I^2^ was greater than 50% or if there was a *p*-value of less than 0.10 [32]. Statistical analyses were performed using the Open Meta-Analyst v.10 software and Metaanal macro for SAS Statistics 9.4M7 software for Windows (SAS Ins., Rayleigh, NC, USA) available at https://www.hsph.harvard.edu/donna-spiegelman/software/ accessed on 8 September 2022.

## 3. Results

### 3.1. Process of Study Selection and Features

The search strategy resulted in a total of 197 records. Abstract and title screening led to the inclusion, in the first phase, of 65 studies. As a result of the second stage, full-text reading led to the exclusion of 40 studies; in the end, 25 studies were included (Figure 2). 

Excellent agreement between reviewers was shown by a κ-statistic value of 0.87. A total of 2553 patients was included in the final meta-analysis of the 25 articles included in this systematic review [6,33,34,35,36,37,38,39,40,41,42,43,44,45,46,47,48,49,50,51,52,53,54,55,56,57]. The collected descriptive characteristics of the chronologically included studies are summarized in Table 1.

Patients were recruited from 1983 to 2011 and the publication year ranged from 2002 to 2020 [33,56]. Sample sizes ranged between 45 and 230 patients [47,52] and originated from 11 different countries across Europe, North America, and Asia. Cytoplasmic E-cad expression was found in the cell membrane in all the studies. The cut-off points for E-cad expression changed across studies, although 50% was the most frequent [33,36,39,41,45,53,54]. Various anti-E-cad antibodies were used, most commonly Clone 36 [6,38,40,41,43,46,52], NCH-38 [29,30,31,45,51], and sc-8426 monoclonal antibodies [37,40,53,54,57]. Table 2 displays the data synthesis for the pooled analysis and its adjustments for each individual citation.

### 3.2. Quality Assessment

Six studies met all REMARK requirements in full. Setting a cut-off point score of >4, 12 studies were considered as low risk of bias, despite 13 studies not fulfilling the REMARK items and these were considered as high risk of bias, as displayed in Table 3. Moderate biases regarding “clinical data” and “immunohistochemistry” domains were evident in some studies because of lacking inclusion criteria information, assessment methodology of E-cad, and study settings. Regarding the “statistics” domain, several studies reported a remarkable risk of biases for inappropriate statistics, erroneous/dubious data reporting, or the absence of appropriate adjustment for confounding factors.

### 3.3. Quantitative Evaluation (Meta-Analysis)

Based on the presence or absence of significant heterogeneity as indicated by the *p*-values from their respective Q tests, the fixed-effect or random-effect models were used to calculate the pooled HR with 95% CI for OS, DSS, and DFS (Figure 3). Although there was some heterogeneity (I^2^ = 59.53%), the random-effects pooled HR value (95% CI) of OS associated with E-cad IHC expression in the tissue of OSCC patients was 0.41 (95% CI 0.32–0.54; *p* < 0.001). Additionally, the DFS fixed-effects pooled HR value was 0.47 (95% CI 0.37–0.61; *p* < 0.001); I^2^ = 0% indicated that there was very little study-to-study heterogeneity (Figure 3B). A fixed-effect meta-analysis of the DSS data revealed a HR of 0.55 (95% CI 0.39–0.76; *p* < 0.001) and no between-study heterogeneity (I^2^ = 0%). Patients with higher E-cad protein expression improved their long-term outcomes, relating to a higher probability of survival and a decreased risk of relapsing the disease.

Table 4 collects the subgroup analysis for all of the 25 studies included in the meta-analysis and their pooled effect estimates. The statistically significant difference was preserved among all subgroups, regarding the OS analysis. Of interest, lower pooled HR emerged in the tongue subgroup (HR = 0.28, 95% CI 0.19–0.43; *p* < 0.001), in comparison with the mixed subsites. Similarly, the investigation of the C36 antibody resulted in a decreased HR (HR = 0.29, 95% CI 0.19–0.43; *p* < 0.001). The statistically significant difference was also preserved for DFS, in all the subgroups; the use of other antibodies rather than clone 35 was considered as a source of heterogeneity (I^2^ = 15.18).

## 4. Discussion

In the present study, we show that tissue E-cad expression can serve as a prognostic biomarker in OSCC, both for OS HR = 0.41 (95% CI 0.32–0.54; *p* < 0.001) and DFS HR = 0.47 (95% CI 0.37–0.61; *p* < 0.001)

Intercellular adhesion plays a major structural role in epithelial integrity and impairment of cohesion between cells, and the altered expression of adhesion molecules in OSCC determines an invasive behavior [58,59]. E-cad is associated with EMT in oral carcinogenesis and this feature strongly defines the prognosis of this neoplasm. The ability to invade adjacent tissues and metastasize locoregionally is one of the most common clinical features in OSCC [40,60]. Multiple cellular events, including changes in the cytoskeleton, matrix protein proteolysis, disruption of cell-to-cell adhesive contacts, and migration, are required for invasion and metastasis processes to occur [20,21]. Crucial for the metastatic ability of cancer are a series of biomolecular changes and events, starting with the intercellular adhesion dysfunctions. E-cad is an invasion suppressor molecule, which means that its loss allows or increases the process of invasion of adjacent normal tissues, due to its role in epithelial intercellular adhesion and in normal tissue morphogenesis [36,42]. In our study, it was demonstrated that low E-cad expression is involved in worse OS, DFS, and DSS in patients with OSCC. Particularly, the HR value of OS was 0.41 with a moderate heterogeneity rate (I^2^ = 59.53%) and in DFS, of 0.47 and in DSS, of 0.55, both with negligible heterogeneity. 

The prognosis of OSCC depends on several factors such as tumor size, degree of differentiation, and stage of disease [38]. Involvement of lymphatic metastasis is present in 30–40% of cases, being one of the main prognostic markers used in the clinic [55]. However, extensive local invasion and/or lymph node metastases are often present at the initial diagnosis, resulting in a more unpredictable prognosis for this type of tumor and consequently, raising doubts about the reliability of these clinical prognostic markers at the time of diagnosis [53]. The primary OSCC that harbors metastases also carries genetic alterations that favor this entire metastatic process, thus being a possible genetic portrait in the prediction of OSCC cases with a high probability of leading to early metastases [47].

Currently, treatment decisions for OSCC are managed through clinicopathological factors such as age, race, sex, histological grade, and tumor/nodule metastasis stage. Obviously, all these factors are valid, but they do not provide accurate information about the general aggressiveness of the tumor and its recurrence potential, demonstrating the importance of indagating new methods to increase the reliability of treatment determination. Therefore, a series of biomarkers are being identified and studied to provide more accurate prognostic information about OSSC, thereby facilitating the management of the neoplasm [40]. A greater understanding about the molecular aspects of OSCC development and progression is needed and the discovery of new targets is of utmost importance [54]. 

The unfavorable prognosis associated with the low expression of E-cad may be a clinically reliable tumor biomarker together with other clinicopathological variables in the decision of the treatment to be used in patients affected by this disease, such as a greater margin of surgical intervention or a combination with radiotherapy/chemotherapy [33]. The upregulation of E-cad may also serve as a new antitumor target strategy, where a stimulation of these cells would occur in order that through increased E-cad expression, a change to a benign behavior in these tumor cells could occur [44]. 

From this point of view, the investigation of the distribution of cadherins and the consequent behavior of tumors related to this expression is a very promising area of interest, as they are biological markers that are cornerstones in predicting the outcome of treatment [47]. Cell adhesion molecules in general, to which cadherins and catenins belong, have a crucial role described in the literature—not only in cell-cell unions but also in cell-extracellular matrix unions. E-cad is the main cadherin of epithelial cells and its role in adherent junctions is responsible for establishing cell–cell contacts [38]. Loss of heterozygosity and inactivating gene mutations, epigenetic silencing at the cancer site, transcriptional repression, cadherin replacement, endocytosis, and proteolytic processing are some of the mechanisms correlated with E-cad inhibition [52].

E-cad has its mechanism of action related to anchoring to the cytoskeleton via β-catenin, which is a cytoplasmic plaque protein responsible for maintaining cell–cell adhesion in normal oral squamous epithelium [58]. Interestingly, the loss of membranous E-cad/β-catenin has been commonly associated with OSCC aggressive behavior through increased tumor invasion, nodal metastasis, and an advanced clinical stage. Multivariate logistic regression analyses demonstrated a loss of membranous expression of E-cad, being the most significant phenotype associated with malignancy, leading to the finding that loss of membranous E-cad/β-catenin are hallmarks of EMT [57]. The EMT process includes, in addition to low intercellular adhesion, the loss of polarity of epithelial cells and increased cell motility, playing a vital role in the progression of cancer [35,54]. This event requires the coordinated expression of several sets of genes and signaling pathways [50]. In EMT, E-cad is suppressed by the expression of Snail, the transcriptional repressor of E-cad, and carcinoma cells that comprise tumor buds end up losing their contact and becoming cellularized [45]. It is likely that several EMT patterns in tumor tissue serve as predictive biomarkers for OSCC progression, which requires further studies, given that OSCC is a pathological entity that remains poorly defined at the molecular level [46,57].

While this systematic review is not the first to be conducted on this topic, previous reviews had several drawbacks. This current meta-analysis is superior as it maintains a precise focus on the expression of E-cad in OSCC, without excluding groups and faithfully considering the correlation of OS, DFS, and DSS with the expression of E-cad. Our study may be more reliable due to lower heterogeneity across studies and the accumulation of studies with a low risk of bias.

Despite the interesting associations that emerged from this systematic review and meta-analysis, results should be read with caution, and they are not free from limitations. Some authors have mentioned the presence of molecular heterogeneities of tumor cells in relation to different tumor locations (e.g., center/superficial portions, invasive tumor fronts) [36,42]. The analysis of the EMT-generated tumor cell buds was performed by one of the authors, who described that positive signals in the nucleus of these tumor buds can impact the accuracy of E-cad expression assessments, which could be one of the reasons for the discrepancies in previous studies [49]. 

Another possible limitation concerns the fact that some studies used different antibodies and different methods (cut-off scoring) to assess staining within the same sample, which is known to affect the results. E-cad levels have also been reported to vary among patients based on ethno-geographical distribution, which could contribute in the inconsistent results shared among studies conducted in different parts of the world [49]. The anatomical location of the tumors was also mentioned as an intervening variable. Differences of distribution/staining of E-cad expression exist according to the site of the primary tumor, which in the case of OSCC, is predominant as membrane staining. This is especially particular, since the larynx usually exhibits both membrane and cytoplasmatic staining, while in the oropharynx, it is almost exclusively cytoplasmic [49]. There was an overall consistency with previous data, although it should be borne in mind that our results showed a stronger biomarker performance for E-cad under the features outlined in our subgroup analysis.

## 5. Conclusions

E-cad appears to be a potential indicator of increased invasiveness and tumor metastasis associated with poor outcomes in OSCC. E-cad regulates complex mechanisms impacting epithelial tumor cell differentiation and epithelial–mesenchymal transitions with translational pathways, translating biomolecular events into clinical outcomes.

If used in conjunction with other clinical and histological indicators, the low expression of E-cad may be a reliable clinically-useful tumor marker in the decision-making of the treatment to be used in these patients with anti-E-cad regimens, which may support surgical resections and improve patient survival. In addition, new antitumor therapeutic strategies and drugs may arise from the upregulation of these classic cadherins, which could have a huge impact on improving the prognosis for OSCC. Forthcoming studies with well-designed inclusion criteria and larger sample sizes evaluating the association of low E-cad expression are still needed to validate the use of this protein as a tumor marker in OSCC.

## Figures and Tables

**Figure 1 biology-12-00239-f001:**
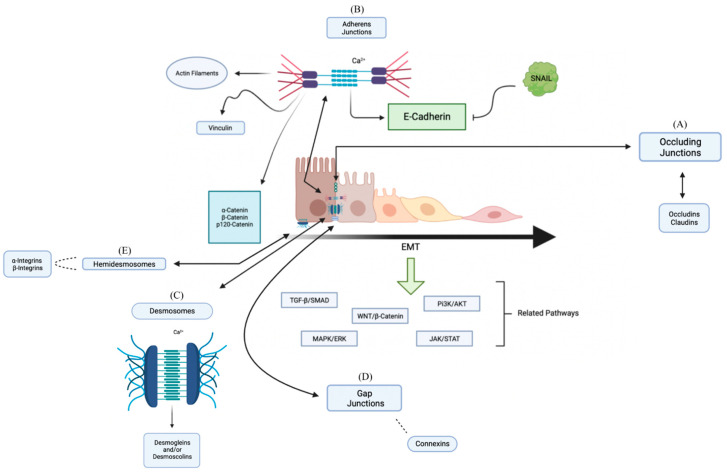
The descriptive diagram model of composition and interactions between proteins that are part of the cell union complex. (**A**) The Occlusion Junctions seal the spaces between epithelial cells, and it is of paramount importance in sealing the epithelial tissues to water molecules and ions, making the passive diffusion of ions impossible; here Occludins and Claudins act. (**B**) The Adherens Junctions connect the bundles of Actin Filaments of one cell with the bundles of another cell where the primary function is to promote adhesion between neighboring cells being crucial in tissue architecture; here Cadherins, α-Catenins, β-Catenins, p120-Catenin, and Vinculins acts. The regulation of the cell–cell adhesion complex is mediated by E-Cadherin and can favor the transformation of epithelial cells into mesenchymal cells after triggering EMT inducers in the various pathways involved; among those involved, the SNAIL protein helps to promote this process by blocking the E-Cadherin. (**C**) Desmosomes connect the Intermediate Filaments of one cell with those of the other cell, with the main aim of promoting intercellular adhesion, not interfering with other junctions; here Desmogleins, Desmocolins, Plakoglobin, Desmoplakin, and the Intermediate Filaments are involved. (**D**) The Gap Junctions allow the passage of water-soluble molecules from one cell to another; it is formed by six subunits of Connexins forming a hexagonal complex. (**E**) Hemidesmosomes anchor the cell’s Intermediate Filaments to the Extracellular Matrix—Cell/Matrix adhesion anchors the Actin Filaments of the cell to the Extracellular Matrix, connecting the Intermediate Filaments of the Cytoskeleton with the Extracellular Matrix; here α-Integrins and β-Integrins are involved.

**Figure 2 biology-12-00239-f002:**
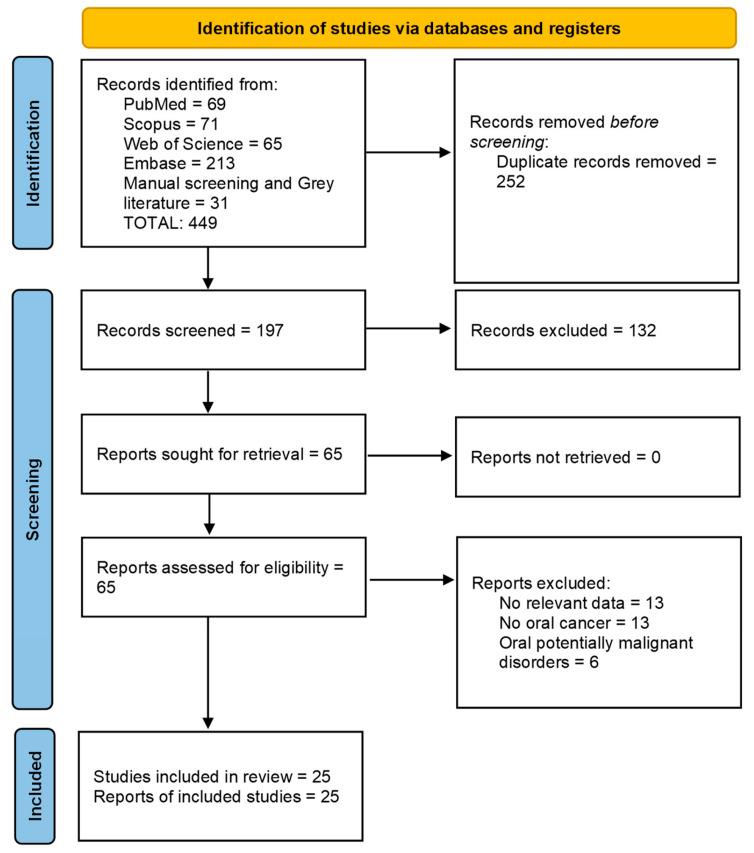
PRISMA 2020 flow diagram.

**Figure 3 biology-12-00239-f003:**
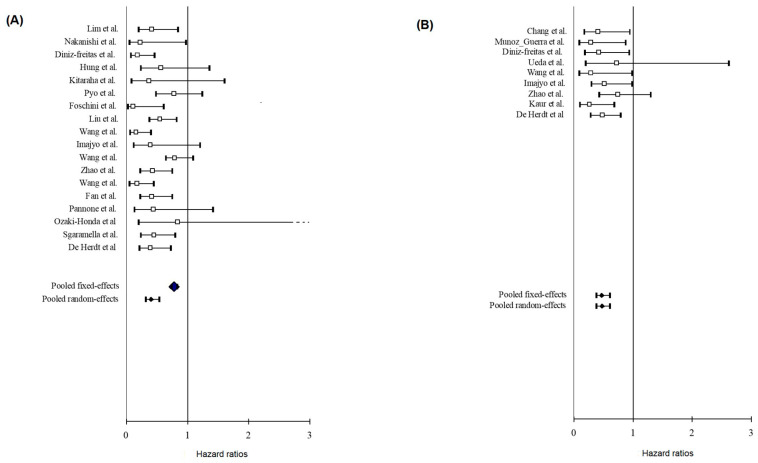
(**A**) Forest plots of overall survival and (**B**) disease-free survival. Horizontal axis represents Hazard Ratios.

**Table 1 biology-12-00239-t001:** General characteristics of included studies.

Study	Year	Country	Sample Size	Tumor Subsite	Recruitment Period	Follow-Up (Months)	E-Cadherin Antibody	Cut-Off Point (%)	E-Cadherin (+) Cases
Chang et al.	2002	China	109	Tongue	N/A	63	H5250 (monoclonal)	50	18
Lim et al.	2004	Japan	56	Tongue	1992–2000	24	Clone 36 (monoclonal)	50	18
Nakanishi et al.	2004	Japan	91	Tongue	1983–1995	58	HECD-1 (monoclonal)	80	28
Munoz-Guerra et al.	2005	Spain	50	Tongue, floor of mouth	1987–2000	36	Clone 36 (monoclonal)	NA	28
Diniz-freitas et al.	2006	Spain	47	Tongue, floor of mouth, alveolar ridge, retromolar trigone, palate, buccal mucosa	1995–2000	36	Clone 36 (monoclonal)	10	33
Hung et al.	2006	Taiwan	45	Tongue, floor of the mouth, gingiva, palate	1999–2005	29	sc-8426 (monoclonal)	10	19
Ueda et al.	2006	Japan	135	NA	1990–1999	68	Clone 36 (monoclonal)	90	54
Kitaraha et al.	2007	Japan	80	Tongue, floor of the mouth, gingiva, buccal mucosa, palate, lip	1990–2005	NA	Clone 36 (monoclonal)	70	34
Pyo et al.	2007	USA	49	NA	1992–1999	NA	HECD-1 (monoclonal)	75	10
Forischini et al.	2008	Italy	58	NA	N/A	19.7	NCH-38 (monoclonal)	50	20
Wang et al.	2009	China	52	Tongue, gingiva, floor of the mouth, buccal mucosa, palate	1994–2001	38.8	589 (monoclonal)	11	30
Liu et al.	2010	China	83	Tongue, gingiva, floor of the mouth, buccal mucosa, palate, lip	1994–2004	50.1	ZM-0092 (monoclonal)	80	31
Wang et al.	2011	China	230	Tongue	1996–2005	65	Clone 36 (monoclonal)	90	30
Imajyo et al.	2012	Japan	152	Tongue, floor of the mouth floor, buccal mucosa, gingival	1993–2006	NA	Clone 36 (monoclonal)	NA	104
Rosado et al.	2012	Spain	59	Tongue, floor of the mouth, gingiva, buccal mucosa, palate, lip	1990–1992	55	HECD-1 (monoclonal)	67	37
Wang et al.	2012	China	76	Tongue	1996–2005	65	sc-8426 (monoclonal)	50	34
Zhao et al.	2012	China	98	Tongue, floor of the mouth floor, buccal mucosa, gingival	2001–2003	60	sc-8426 (monoclonal)	40	49
Kaur et al.	2013	India	105	Tongue, floor of the mouth, lip, gingiva, hard palate, soft palate, retromolar trigone and floor of the mouth	2002–2005	24	sc-8426 (monoclonal)	50	61
Wang et al.	2013	China	67	Tongue	1996–2005	65	sc-8426 (monoclonal)	50	27
Fan et al.	2014	Taiwan	74	Oral mucosa, Buccal, Gingival, palate, Other	1999–2006	NA	NCH-38 (monoclonal)	50	22
Pannone et al.	2014	Italy	164	Tongue, floor of the mouth, buccal mucosa, retromolar trigone, gingiva, palate, lip	1990–2006	39,41	Clone 36 (monoclonal)	NA	152
Da Silva et al.	2015	Brazil	102	NA	N/A	120	NCH-38 (monoclonal)	5	68
Ozaki-Honda et al	2017	Japan	76	NA	1983–2002	NA	Ab40772 (monoclonal)	50	55
Sgaramella et al.	2018	Sweden	120	Tongue	N/A	47	M3612 (monoclonal)	20	118
De Herdt et al	2020	The Netherlands	203	Tongue, floor of mouth, gingiva, palate	1984–2010	NA	NCH-38 (monoclonal)	50	NA
Wangmo et al.	2020	Thailand	200	Tongue, Floor of the mouth, gingiva, buccal mucosa	2008–2011	NA	NCH-38 (monoclonal)	60	172

**Table 2 biology-12-00239-t002:** Data synthesis for pooled analysis and adjustments for each included study.

Study	Year	Overall Survival (HR 95% CI)	Disease-Free Survival/Progression Free Survival (HR 95% CI)	Disease-Specific Survival (HR 95% CI)	Adjustment
Chang et al.	2002		0.40 (0.17–0.94)		Cox multivariate regression adjusted for gender, grade, TNM stage, T stage and N status.
Lim et al.	2004	0.41 (0.20–0.84)			Cox multivariate regression adjusted for stage, location, and N status.
Nakanishi et al.	2004	0.22 (0.05–0.97)			Cox multivariate regression adjusted for age, gender, tumor differentiation, N status, TNM stage, growth pattern, invasion depth, and vessel invasion.
Munoz-Guerra et al.	2005		0.28 (0.09–0.87)		Univariate.
Diniz-freitas et al.	2006	0.18 (0.07–0.46)	0.41 (0.18–0.93)		Cox multivariate regression adjusted for stage and surgical margin status.
Hung et al.	2006	0.56 (0.23–1.36)			Univariate.
Ueda et al.	2006		0.71 (0.19–2.65)		Cox multivariate regression adjusted for tumor differentiation, T status, N status, TNM stage, and mode of invasion.
Kitaraha et al.	2007	0.36 (0.08–1.60)			Univariate.
Pyo et al.	2007	0.77 (0.48–1.24)			Cox multivariate regression adjusted for tumor differentiation, T status, N status, TNM stage, and mode of invasion.
Foschini et al.	2008	0.11 (0.02–0.61)			Univariate.
Wang et al.	2009		0.28 (0.08–0.98)		Univariate.
Liu et al.	2010	0.55 (0.37–0.82)			Univariate.
Wang et al.	2011	0.15 (0.06–0.40)			Univariate.
Imajyo et al.	2012	0.38 (0.12–1.20)	0.51 (0.29–0.98)		Univariate.
Rosado et al.	2012			0.56 (0.34–0.92)	Cox multivariate regression adjusted for TNM stage, T stage and N status.
Wang et al.	2012	0.78 (0.64–1.09)			Univariate.
Zhao et al.	2012	0.42 (0.22–0.75)	0.74 (0.42–1.30)		Cox multivariate regression adjusted for TNM stage and N status.
Kaur et al.	2013		0.26 (0.10–0.68)		Univariate.
Wang et al.	2013	0.15 (0.05–0.45)			Univariate.
Fan et al.	2014	0.41 (0.22–0.75)			Cox multivariate regression adjusted for betel quid chewing, cigarette smoking, tumor size, TNM stage, and recurrence.
Pannone et al.	2014	0.43 (0.13–1.42)			Univariate.
Da Silva et al.	2015			0.48 (0.23–1.00)	Univariate.
Ozaki-Honda et al	2017	0.83 (0.20–3.48)			Univariate.
Sgaramella et al.	2018	0.45 (0.23–0.80)			Univariate.
De Herdt et al	2020	0.39 (0.21–0.72)	0.47 (0.28–0.79)		Univariate.
Wangmo et al.	2020			0.57 (0.34–0.96)	Cox multivariate regression adjusted for T stage, N status, clinical stage and treatment.

**Table 3 biology-12-00239-t003:** Risk of bias assessment of included studies based on REMARK evaluation items.

Study	Year	Samples	Clinical Data	Immunohistochemistry	Prognostication	Statistics	Classical Prognostic Factors	Overall
Chang et al.	2002	I	I	I	A	A	A	3
Lim et al.	2004	A	A	I	A	A	A	5
Nakanishi et al.	2004	A	A	A	A	A	A	6
Munoz-Guerra et al.	2005	A	I	A	I	I	A	3
Diniz-freitas et al.	2006	A	A	A	A	A	A	6
Hung et al.	2006	A	A	A	A	I	I	4
Ueda et al.	2006	A	A	A	A	A	A	6
Kitaraha et al.	2007	A	A	A	A	I	A	5
Pyo et al.	2007	A	I	A	I	A	A	5
Foschini et al.	2008	A	I	I	I	I	I	1
Wang et al.	2009	A	A	I	I	I	I	2
Liu et al.	2010	I	A	I	A	I	I	2
Wang et al.	2011	A	A	I	A	I	I	3
Imajyo et al.	2012	A	A	A	A	I	A	5
Rosado et al.	2012	A	I	A	I	A	A	4
Wang et al.	2012	A	I	I	A	I	A	3
Zhao et al.	2012	A	A	A	A	A	A	6
Kaur et al.	2013	I	I	I	I	I	A	1
Wang et al.	2013	A	I	A	A	I	A	4
Fan et al.	2014	A	A	A	A	A	A	6
Pannone et al.	2014	A	A	A	A	I	I	4
Da Silva et al.	2015	A	A	A	A	I	A	5
Ozaki-Honda et al	2017	A	I	I	I	I	I	1
Sgaramella et al.	2018	I	A	A	I	I	I	2
De Herdt et al	2020	A	A	A	A	I	A	5
Wang Mo et al.	2020	A	A	A	A	A	A	6

**Table 4 biology-12-00239-t004:** Subgroups pooled hazard ratios and 95% confidence intervals for long-term outcomes.

		Number of Studies (*n*)	Pooled HR (95% CI), Fixed Effects	Pooled HR (95% CI), Random Effects	*p* Value	I^2^ (%)	Q Test *p* Value
Overall survival							
	Overall	18	0.78 (0.74–0.83)	0.41 (0.32–0.54)	<0.001	59.52	<0.001
	High quality	9	0.45 (0.35–0.57)	0.43 (0.32–0.57)	<0.001	20.30	0.26
	Low quality	9	0.80 (0.76–0.85)	0.51 (0.37–0.69)	<0.001	77.61	<0.001
	Full adjustment	6	0.47 (0.35–0.62)	0.42 (0.28–0.63)	<0.001	47.94	0.09
	Asian	12	0.80 (0.75–0.84)	0.48 (0.37–0.63)	<0.001	75.27	<0.001
	Non-Asian	6	0.47 (0.35–0.64)	0.39 (0.24–0.65)	<0.001	55.87	<0.001
	Tongue	5	0.28 (0.19–0.43)	0.28 (0.17–0.45)	<0.001	29.34	0.23
	Mixed subsites	13	0.79 (0.75–0.84)	0.55 (0.43–0.69)	<0.001	68.03	<0.001
	Use of Clone 36 antibody	5	0.29 (0.19–0.43)	0.29 (0.19–0.43)	<0.001	0	0.47
	Use of other antibodies	13	0.79 (0.75–0.84)	0.54 (0.43–0.69)	<0.001	71.24	<0.001
	Use of 50% cut-off point	7	0.81 (0.76–0.85)	0.52 (0.38–0.72)	<0.001	78.96	<0.001
	Use of other cut-off points	11	0.47 (0.38–0.58)	0.42 (0.31–0.57)	<0.001	38.60	0.09
Disease-free survival							
	Overall	9	0.47 (0.37–0.61)	0.47 (0.37–0.61)	<0.001	0	0.64
	High quality	5	0.54 (0.41–0.72)	0.54 (0.41–0.72)	<0.001	0	0.72
	Low quality	4	0.31 (0.19–0.52)	0.31 (0.19–0.52)	0.004	0	0.91
	Full adjustment	4	0.55 (0.39–0.76)	0.55 (0.39–0.76)	<0.001	0	0.68
	Use of Clone 36 antibody	4	0.46 (0.31–0.69)	0.46 (0.31–0.69)	<0.001	0	0.72
	Use of other antibodies	5	0.48 (0.36–0.66)	0.47 (0.33–0.67)	<0.001	15.18	0.32
Disease-specific survival							
	Overall	3	0.55 (0.39–0.76)	0.55 (0.39–0.76)	<0.001	0	0.92

## Data Availability

Data can be requested from the corresponding author upon justified query.

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
