# Peer review of "Overexpression of E-Cadherin Is a Favorable Prognostic Biomarker in Oral Squamous Cell Carcinoma: A Systematic Review and Meta-Analysis"

_biology, 2023, doi:10.3390/biology12020239_

Round 1

Reviewer 1 Report

Although this review include adequate sample size, the cut-off point for each study varies widely to summarize these population  for the meta-analysis.

A analysis of the studies based on Asia and Europe-America may be useful.

You should mention that the DFS is not statistically significant and mention the limitations of the study earlier in the discussion.

Author Response

First of all, we wish to thank the reviewer from his/her work and dedication to our article. His/Her useful comments and suggestions really contributed to the improvement of our manuscript.

Although this review includes adequate sample size, the cut-off point for each study varies widely to summarize these population  for the meta-analysis.

We sincerely appreciate this valuable suggestion. It is true that the articles used a wide variety of cut-off points to determine e-cadherin positivity, and that is a relevant limitation. However, this eventuality is frequent in meta-analyses carried out to evaluate the validity of biomarkers based on immunohistochemistry to assess long term outcomes in OSCC, as our group has unravelled in previous research:

Lorenzo-Pouso AI, Gallas-Torreira M, Pérez-Sayáns M, Chamorro-Petronacci CM, Alvarez-Calderon O, Takkouche B, Supuran CT, García-García A. Prognostic value of CAIX expression in oral squamous cell carcinoma: a systematic review and meta-analysis. J Enzyme Inhib Med Chem. 2020 Dec;35(1):1258-1266. doi: 10.1080/14756366.2020.1772250.

Silva FFVE, Padín-Iruegas ME, Caponio VCA, Lorenzo-Pouso AI, Saavedra-Nieves P, Chamorro-Petronacci CM, Suaréz-Peñaranda J, Pérez-Sayáns M. Caspase 3 and Cleaved Caspase 3 Expression in Tumorogenesis and Its Correlations with Prognosis in Head and Neck Cancer: A Systematic Review and Meta-Analysis. Int J Mol Sci. 2022 Oct 8;23(19):11937. doi: 10.3390/ijms231911937.

Troiano G, Caponio VCA, Zhurakivska K, Arena C, Pannone G, Mascitti M, Santarelli A, Lo Muzio L. High PD-L1 expression in the tumour cells did not correlate with poor prognosis of patients suffering for oral squamous cells carcinoma: A meta-analysis of the literature. Cell Prolif. 2019 Mar;52(2):e12537. doi: 10.1111/cpr.12537.

However, we would like to mention one strength of this work in relation to others. This is that in subgroup analysis for the cut-off point of 50% of the stained cells reached a strong association with the overall survival, meaning therefore that the association prevails despite the adjustment for this type of covariates that subgroup analysis can clear (OS_HR50%cut-off= 0.52 (95% CI0.38–0.72).

In any case, this has now been mentioned and added as a limitation at the end of the paper thanks to your wise words.

An analysis of the studies based on Asia and Europe-America may be useful.

Thank you for the suggestion. This was already addressed as subgroup analysis in the Table 4, showing statistically significant association in both Asian and Non-Asian pooled studies (Europe-America).

You should mention that the DFS is not statistically significant and mention the limitations of the study earlier in the discussion.

We sincerely appreciate this valuable comment. We have made a big mistake in indicating these values during the beginning of the discussion, because we announced some values for the value of the Q test for heterogeneity as if it were the p value of the tests based on the meta-analysis computations. All pooled computations as well as subgroups that were performed in the analysis were significant and therefore were exposed. We now make amend our mistake and sincerely thanks your valuable comment.

Once again, we would like to express our gratitude for your constructive criticism.

Reviewer 2 Report

It is important to identify biomarkers for Oral squamous cell carcinoma (OSCC), and E-cadherin family proteins play a vital role. The manuscript is well written, however, certain information need to be updated from further consideration: 

1. Introduction: E-cadherin family has divisions, such as desmosomes, and also subdivisions such as desmogleins. I recommend adding the classification of E-cadherin junctional proteins and a figure would be appropriate in the introduction. 

I also recommend adding two articles in the literature review where it has been shown that E-cadherin (desmoglein 3) overexpression plays an important role in OSCC invasion and migration at the cellular level.

https://doi.org/10.1080/19336918.2016.1195942 

https://doi.org/10.1016/j.yexcr.2018.06.037 

Author Response

It is important to identify biomarkers for Oral squamous cell carcinoma (OSCC), and E-cadherin family proteins play a vital role. The manuscript is well written, however, certain information needs to be up.

First of all, we wish to thank the reviewer from his/her work and dedication to our article. His/Her useful comments and suggestions really contributed to the improvement of our manuscript.

Introduction: E-cadherin family has divisions, such as desmosomes, and also subdivisions such as desmogleins. I recommend adding the classification of E-cadherin junctional proteins and a figure would be appropriate in the introduction. I also recommend adding two articles in the literature review where it has been shown that E-cadherin (desmoglein 3) overexpression plays an important role in OSCC invasion and migration at the cellular level. https://doi.org/10.1080/19336918.2016.1195942 https://doi.org/10.1016/j.yexcr.2018.06.037 

Thank you for the valuable input. We have now included a figure including your valuable input as its corresponding legend briefly discussing your argument.

Once again, we would like to express our gratitude for your constructive criticism.

Round 2

Reviewer 1 Report

-